# Characteristics and Causes of Construction Accidents in a Large-Scale Development Project

**Albert P. C. Chan** [1,2], **Yang Yang** [1,2], **Tracy N. Y. Choi** [2,*] **and Janet Mayowa Nwaogu** [2]

1    Shenzhen Research Institute, Hong Kong Polytechnic University, Shenzhen 518057, China;
     bsachan@polyu.edu.hk (A.P.C.C.); jackie.yyang@polyu.edu.hk (Y.Y.)
2    Department of Building and Real Estate, Hong Kong Polytechnic University, Hong Kong 999077, China;
     janetmayowa.nwaogu@polyu.edu.hk
*    Correspondence: bsnychoi@polyu.edu.hk; Tel.: +852-2766-4305

**Abstract:** The airport at the Kai Tak district was relocated to Chek Lap Kok in 1998. The Hong Kong Government endeavors to redevelop Kai Tak into a mixed-use community. A total of eight fatal construction accidents have happened since the Kai Tak Development (KTD) commenced in 2013, and seven of them occurred between 2020 and 2021. The alarming figures call for an in-depth investigation of fatal accidents, hence the aim of the current research. Detailed accident investigation reports that outlined accident causation and preventive measures for the eight accidents were collected. With a detailed examination of validity and reliability, the modified loss causation model was applied to analyze the situational variables, incident sequences, and causes of accidents. The results showed that "fall" and "struck by" were the most common accidents in the KTD. Several risk factors for fatal accidents were identified, including "aging workers", "new to a construction site", "ethnic minority", "illegal worker", "working on weekends", and "subcontracting companies". Preventive measures were offered to help government authorities and construction practitioners enhance the safety performance of the ongoing KTD projects. This study contributes to the knowledge of construction safety by identifying safety issues of mega interfacing projects. The practice of learning from accidents should be promoted in order to prevent similar accidents from occurring again.

**Keywords:** Kai Tak development; mega interfacing projects; modified loss causation model; accident analysis and prevention

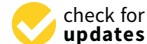

## 1. Introduction

Ignoring safety can create adverse impacts on sustainable development [1] because construction worker injuries do not only affect the safety, health, and quality of life, they also incur direct and indirect costs of construction accidents [2,3]. Although the government authorities, industry practitioners, and researchers put much effort into improving construction safety [4–6], the recurrence of similar types of accidents continues to plague the construction sector worldwide. There were 23,067 accidents related to the fall of a person from height in the US construction industry from 2000 to 2020 [7]. In Hong Kong, fall and struck-by accidents were ranked as the most frequently occurring accidents over the years [8]. Learning from the past has long been an important way to prevent similar accidents from reoccurring [9,10]. However, the recurrence of similar accidents might imply that construction professionals, management staff, and individual workers probably failed to learn lessons from the past [11].

The retrospective analysis of accidents would be helpful to provide helpful information and facilitate learning from the past in the construction sector [12]. Over the last two decades, numerous studies have been carried out to explore the characteristics and causes of similar accidents. For instance, by evaluating the demographics and characteristics of fall fatalities in the US construction industry, Dong et al. [13] found that Hispanic construction

workers were more prone to fall accidents than white, non-Hispanic laborers. Chi et al. [14] summarized nine common causes of 250 fatal electrical accidents in the Taiwan construction industry. Wong et al. [15] applied the Human Factor Analysis and Classification System to explore the root causes of fall fatality in the Hong Kong construction industry. Four common causes were classified: Poor planning, violation, hidden risks caused by others, and incompetent personnel [15]. These studies contributed to the body of knowledge by exploring the common characteristics and causes of similar accidents. Since each building project is unique [16], safety risks associated with specific work environments and construction methods could vary from project to project. Examining construction accidents in the same project would be more beneficial to provide prompt and clear directions to relevant stakeholders involved. For instance, Chan et al. [17] observed five drowning fatalities in a mega public infrastructure project that included land and sea-based construction works. They indicated that construction workers face a high risk of danger when working above or near the sea. Their findings would serve as valuable references for construction practitioners to take necessary safety measures in similar projects worldwide. In view of this, the project-level analysis of construction accidents would be helpful to draw vital lessons for relevant project stakeholders and prevent accidents from happening again in the same project [17,18].

It has been reported that eight fatalities occurred on eight construction sites of the KTD since the projects commenced in 2013, according to the authors' preliminary search of newspaper databases via NewsWise. This database has been used to acquire accident information by previous studies [17,19]. The first fatal accident occurred in 2014. Four accidents happened within two successive months in 2020, accounting for 10% of total fatal accidents in the construction industry [8]. Three occurred within successive eight days in April 2021, which occupied 23% of the fatal construction accidents in the first three quarters of 2021 [8] (As of the date of the manuscript preparation, the data on fatal accidents in the construction industry in the first three quarters of 2021 are available only.). Such an abnormally high frequency of fatal accidents in the KTD calls for an in-depth investigation of the characteristics and causes of these accidents to suggest preventive measures and draw vital lessons for relevant project stakeholders. Hence, this study aimed to investigate the features and causes, and recommend preventive strategies for the construction incidents in the KTD.

*Background of the Kai Tak Development*

Located on the east side of Kowloon Bay, the Kai Tak Airport served as Hong Kong's international airport from 1925 to 1998. Initially, the airport was far away from residential areas, and it was then expanded to accommodate international passenger traffic. The airport became the third busiest airport in the world in 1996 [20]. The nearby residential areas expanded with population growth and increased urbanization as time went by. Consequently, the airport was close to residential areas, resulting in overcrowding, noise, pollution, and safety concerns. Therefore, the Hong Kong Special Administrative Region Government announced in 1998 that the Kai Tak Airport was relocated to Chek Lap Kok on Lantau Island, leaving the obsolete airport in the metro region for significant development.

The Kai Tak Development (KTD) was designed as a large-scale and highly complex development project spanning over 320 hectares, with the aim to develop a mixed-use of commerce, housing, leisure, infrastructure, and community [21]. To attain this goal, mega interfacing projects in and around the KTD have been launched. To be specific, public works projects include public rental housing and a great variety of government and community facilities. The government and community facilities include hospitals, schools, town hall, multi-purpose complexes, indoor recreation centers, a library, a stadium by the Victoria Harbour, a metropolitan park, a walking trail, a marine passenger terminal, and new metro facilities [21]. The development commenced in 2013. As of 2021, some public and private projects have been completed, while others, such as Kai Tak sports park, are still under construction.

## 2. Accident Causation Models

Accident causation analysis is an effective method to identify the root causes of accidents, which serves as a basis for further corrective actions. Different approaches have been suggested to support the identification of factors and causes that contribute to accidents [22]. Traditional approaches to accident investigation tended to provide a simple explanation for accident causation by concentrating mainly on physical events and human errors (e.g., Abdelhamid et al. [23]). Accident causation models then evolved, representing a significant shift from sequential events, single failures, and individual operation errors toward the examination of the complex interaction among organizational factors, management decision-making process, and human actions [24,25]. Systemic models are thus viewed as more comprehensive to offer an in-depth understanding of accident causation by considering political, cultural, financial, and technological influences in complex sociotechnical systems [26]. A brief overview of the systemic models is given in Table 1.

**Table 1.** Overview of systemic models for accident causation analysis.

| Accident Causation Model | Features | References |
|---|---|---|
| AcciMap | Graphically represent how and why an organizational accident occurred throughout the sociotechnical system across government policy and budgeting; regulatory bodies and associations; local area government planning and budgeting (including company management), technical and operational management; physical processes and actor activities; and equipment and surroundings | [27] |
| Systems Theoretic Accident Model and Processes (STAMP) | A constraints-based model, viewing accidents as the result of the inadequate control of safety-related constraints, potentially as comprehensive as AcciMap | [28,29] |
| Functional Resonance Accident Model (FRAM) | The interaction of three components: Organizational, technical, and human | [30] |
| Constraint-response model | A constraints-based model, viewing accidents as the result of constraints and responses experienced by all construction project participants | [31,32] |
| Modified loss causation model (MLCM) | Classify the causes of construction accidents into three categories: Immediate causes, safety management system failures, and underlying factors | [33] |
| Mitropoulos's system model | View errors and accidents as the result of joint effects of task unpredictability, production pressure, and efficient behaviors | [22,34] |
| Loughborough construction accident causation model (ConCa) | Classify the causes of construction accidents into three groups: Immediate circumstances, shaping factors, and originating influences | [35,36] |
| Human factor and classification system (HFACS) | Classify accident causes into four layers: organizational influences, unsafe management, preconditions for unsafe acts, and unsafe acts themselves | [37] |

Several systemic models, such as AcciMap [27], Systems Theoretic Accident Model and Processes (STAMP, [28]), and Functional Resonance Accident Model (FRAM, [30]), have been developed. Despite the ubiquitous advantages of systemic models in research and practice in many industrial settings, their usage in construction has been minimal [38]. Rosa et al. [39] used the FRAM to identify risks associated with the construction industry processes by assessing the complex interactions between workers' behavior and work activity within complex socio-technical systems (i.e., a non-linear approach). The FRAM was also used to determine the critical variabilities and human factors that contributed to the development of safe drilling operations [40]. AcciMap was applied by Zhou et al. [41]

to delineate the system levels and causal paths of the contributory factors to tower crane safety. The key strengths of systemic models lie in that the contributory factors and their interactions are evaluated within the entire work system, ranging from individual workers/equipment to the organization and beyond. Despite these merits, systemic models rely on the data involved to construct the analytic framework. These data are not limited to the accident itself but may include government policy, legislation, rules, and regulations. However, the inaccessible and unavailable data about the entire sociotechnical system may make the systemic models less favorable in construction. For instance, ineffective government policies and regulations are rarely investigated or reported in accident investigation reports. In fact, most construction accident causation models hardly account for the effect of regulatory body and government level factors on accident causation [38].

Many efforts have been taken to devise and implement systemic accident causation models for the construction sectors, although those models cover only as far as the organizational contributing factors [38]. Suraji et al. [31] developed a constraint-response model, indicating that the accidents could result from constraints and responses experienced by all construction project participants. Suraji [32] further applied this model to analyze the contributory factors of 1000 construction accidents. Chua and Goh [33] modified the loss causation model (MLCM) to classify the causes of construction accidents into three categories: Immediate causes, safety management system failures, and underlying factors. This model could help organizations identify deficiencies in the safety management system and organizational culture and implement systematic actions to remove those flaws [33]. The MLCM was used by Chan et al. [17] to analyze the causes of fatal accidents in a large-scale infrastructure project. Mitropoulos et al. [22] proposed a descriptive model that views errors and accidents as the result of joint effects of task unpredictability, production pressure, and inefficient behaviors [34].

Haslam et al. [35] and Harvey et al. [36] developed a construction causal accident model (ConCa), also known as the Loughborough construction accident causation model. Built upon the Reason's Swiss Cheese model, the ConCa model outlines three groups of causes: Immediate circumstances, shaping factors, and originating influences. This model was effective in the causal analysis of construction accidents [12,42]. However, some limitations of the ConCa model were noticed by Cooke and Lingard [42]. First, the factors' categorization was subject to the analyst's interpretation, and thus, different causal pathways might be identified. Second, not all incident scenarios were adequately represented by the hierarchical sequence of the three levels of contributory factors [12,42]. Originating from the Swiss Cheese model, the Human Factor and Classification System (HFACS) classified accident causes into four layers: Organizational influences, unsafe management, preconditions for unsafe acts, and unsafe acts themselves [37]. Since the HFACS taxonomy can be used to determine the failures and weaknesses of the safety system, the HFACS has been successfully applied to provide a retrospective analysis of the root causes of construction accidents [15,43]. Wong et al. [15] modified the HFACS by integrating the environmental factors and hazards created by others into the classification system. Ye et al. [44] improved the HFACS by adding external elements (regulatory and economic aspects, social, political, and legal environment) into the analysis of construction accidents.

Although the aforementioned theoretical models may enable safety researchers and practitioners to conceptualize the mechanism leading to construction accidents, reports of utilizing these models to analyze the actual events are relatively scarce [15]. The application of the theoretical models for analyzing actual accidents faces several challenges. Many underlying contributory factors indicated by the theoretical models are usually unexplored. Hence, the advantages of those models cannot be fully taken for their intended purposes. In addition, subjective bias and human errors of analysts using those models may result in unreliable and inconsistent findings from the models. To overcome these challenges, the present study attempted to determine the appropriate theoretical model to analyze fatal accidents in the KTD. The validity and reliability of the selected model were subsequently

evaluated. The assessment of validity and reliability is recognized as a key step of accident causation modeling [24,45], but has been largely ignored by previous studies [38].

Overall, the study attempts to examine the safety issues of the KTD mega interfacing projects. In terms of research contributions, the present study would contribute to the knowledge of construction safety by exploring the safety performance and safety issues of mega interfacing projects. The study would adopt a construct research methodology by selecting appropriate accident causation models and examining the validity and reliability of the model used. This is one of the first studies to adopt this methodology in construction safety research to the authors' knowledge. The findings of the accident causation analysis would be helpful to draw important lessons for government authorities, clients, main contractors, subcontractors, and frontline workers involved. Necessary and remedial preventive measures can be adopted promptly to prevent similar accidents in the ongoing KTD projects.

## 3. Methods

### 3.1. Data Collection and Processing

The research methodology of this study is depicted in Figure 1. The Coroner's Court, one of the court services of the Judiciary of Hong Kong SAR, provided the dossier for each case, which included the summary conclusions at the inquest, the accident investigation report documented by the Labour Department, the police report, etc. The death investigation report is the record of the coroner's inquest into a case, supported by testifications from field experts and witnesses. The objective of the inquest is to analyze all the evidence collected and objectively inquire actual causes and facts of an accident and the situations surrounding the event, rather than pointing to the fault, responsibility, or compensation [15]. The accident investigation report carried out by the Labour Department provided information about the decedent, contract employment situations, background, findings, inspection, judgments, and preventive recommendations. The police report included the statement given by witnesses concerning the accident and the summary report of the police investigation. The above information is viewed as a reliable accident causation analysis report tested by law, supported by evidence, and witnessed by all stakeholders involved in the case [15].

### 3.2. Descriptive Analysis

Each case was reviewed by extracting descriptive information about the decedents and the accidents. Specifically, demographic information (e.g., gender, work trade, age, length of working experience) of decedents was first presented. Features of each accident, such as date and time of the event, type of project, and type of employer, were also outlined.

### 3.3. Selection of Appropriate Causation Model

Adopting an appropriate causation model would guide researchers and practitioners to identify the root causes of construction accidents, thereby implementing necessary corrective actions to prevent the recurrence of similar accidents in the future. The selection of an appropriate causation model relies on two criteria: Scientific rigor and practicality. While the scientific rigor of the constraint-response model, MLCM, ConCa, Mitropoulos's model, and HFACS has been addressed above, the application of these models to actual construction accidents remains scarce. Therefore, a pilot trial was conducted to determine which models could meet the designated criteria while explaining accident causes. Case 4 was randomly selected for the pilot trial.

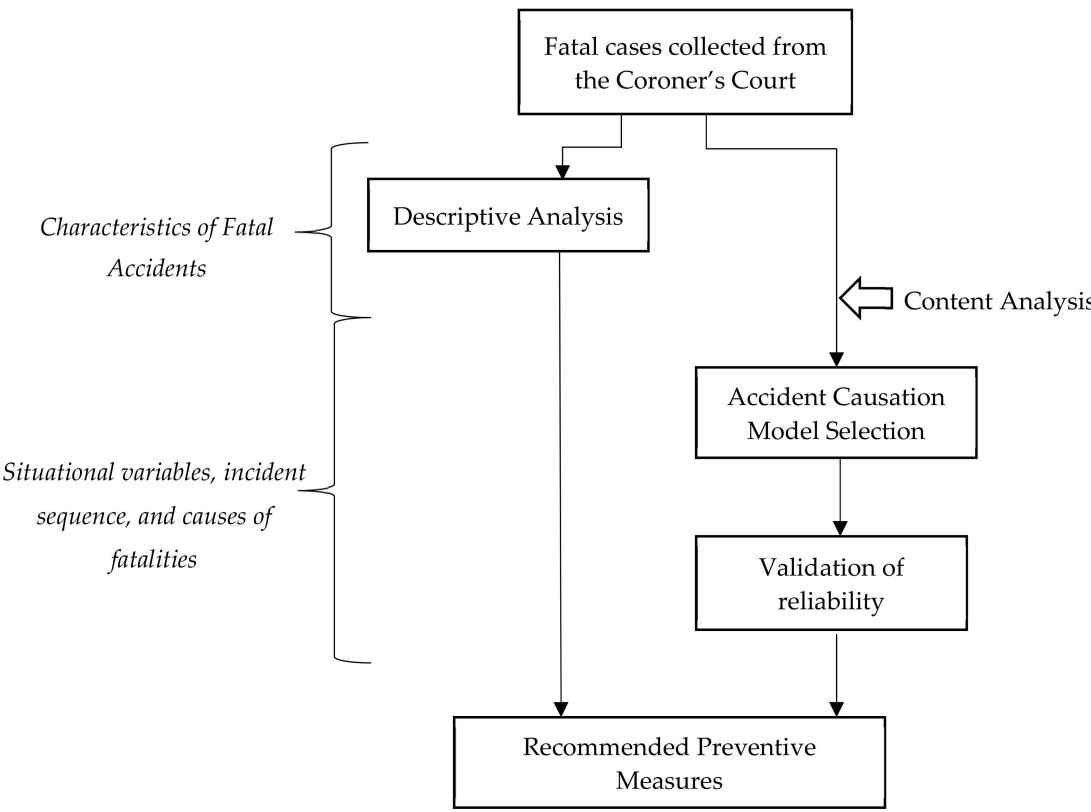

**Figure 1.** Research methodology.

A content analysis was performed to convert unstructured accident data into a format that can be analyzed [46]. The content analysis enables researchers to examine data precisely and give codes to paragraphs or sections. Through the content analysis, the texts of reports were coded by the taxonomies implied by each accident causation model. For instance, the accident causes were classified into the hierarchical HFACS taxonomies, unsafe acts, preconditions for unsafe acts, unsafe supervision, and organizational influences. Similar exercises were performed for the constraint-response model, MLCM, ConCa, and Mitropoulos's model. The findings of the causal analysis using different causation models are shown in Appendix A. Some contributory factors indicated by the ConCa, constraint-response, and Mitropoulos's models were unknown based on the available information. The hierarchical sequence of the three levels of contributing elements in the HFACS model did not sufficiently capture all event circumstances. For instance, in Case 4, the fatal accident was caused by the hazards created by others, while unsafe acts of the decedent were not observed. Both ConCa and MLCM were found comprehensive to address the direct and root causes. The MLCM was selected because it fully interpreted incident sequence and situational variables absent in the ConCa. The situational variables and incident sequence considered by the MLCM would provide a deeper understanding of accident causation. Therefore, the MLCM was finally used to systematically classify the contributory components in each case.

*3.4. Identification of Root Causes by Applying the Selected Model*

According to the MLCM taxonomy, the situational factors and causes of fatal accidents were determined by content analysis of accident reports. To be specific, situational variables included type of accidents, working surface, and type of job and agent involved. Sub-standard or unsafe conditions, substandard or unsafe behaviors, and immediate personal factors are all immediate causes. Lack of safety measures, inadequate safety measures, and inadequate execution are all examples of safety management system failures. Personal, job,

and organizational factors are the three types of underlying factors. Workers' knowledge, experience, skills, and capacity are linked to underlying personal variables, while work design, execution, and supervision are underlying job factors. Organizational factors include safety climate, structure, policies, and culture of the organization. In addition, Chua and Goh [33] further differentiated between immediate personal factors and underlying personal factors. For instance, "improper motivation to save time and effort" was considered an immediate personal factor, while "lack of experience and knowledge" was an underlying personal factor.

### 3.5. Validity and Reliability

The validity and reliability of the findings derived from the MLCM were subsequently examined. According to Branford [47], there are two ways to consider the validity of accident analysis methods. The first approach examines the method's validity by determining whether the method used is suitable for its intended aims and proposes [47]. In this case, the MCLM adopts a system approach with mapping cause-effect relations and provides detailed taxonomies to classify contributory factors. In theory, this model was created to offer insights into how and why a construction accident happens throughout the organizational structure and ensure its internal validity. The second approach to considering the validity is to focus on the validity of the outcomes achieved when the method is used [47]. Four approaches have been recommended to evaluate the validity of the results of accident analysis methods [38,47–51]. First, the validity can be determined by comparing the similarities and differences between the results and a "gold standard". However, this validity evaluation is not practical to use because a "gold standard" is rarely obtainable [48]. The second approach is to evaluate the results in terms of their internal logic. Nevertheless, the internal logic of the results does not necessarily guarantee that a correct or comprehensive answer is obtained [48]. The third approach is to evaluate the degree of similarity between findings generated using various methods. This approach is not applicable to the present study because the other causation models were considered less comprehensive than the MLCM. As a result, finding another "appropriate" causation model to verify the findings of the MLCM becomes unrealistic. The last method is to evaluate the results against those of experts, which was adopted in this study. The MLCM results were verified by a safety professional (lecturer with research experience of over six years in construction safety) and a safety practitioner (senior manager with work experience of over 20 years in construction safety).

Two approaches can be used to assess the reliability of accident analysis methods [38,48,52]. First, the intra-analyst agreement can be examined by how a single analyst using the method produces similar results at different times or analyzes different cases with similar characteristics. Second, the inter-analyst agreement can be assessed by the extent to which the outcomes are consistent regardless of the analyst. In this study, both approaches were adopted. Specifically, two analysts analyzed the eight cases (including four fall accidents and four struck-by accidents) at different times. Their results were cross-checked to examine the consistency of the results after applying the MLCM method.

### 3.6. Classification of Preventive Measures

Precautionary measures for each fatal accident recommended by the Coroner's Court reports were categorized into three major types: Behavioral, engineering, and administrative controls [53]. As stated by Goldenhar and Schulte [53], behavioral controls are an "attempt to influence workers' points of view, understanding, attitudes, and reactions related to hazardous circumstances". Engineering controls represent "engineered or physical manipulations of sources or routes of exposure to occupational hazards." Administrative controls are the "management initiatives that modify a worker's work process and/or work exposure."

## 4. Results and Discussion

### 4.1. Characteristics of Fatal Accidents

Eight fatal construction accidents occurred between 2014 and 2021 in the KTD. Shockingly, four fatalities occurred within two successive months of 2020, while three fatal accidents happened within eight days in April 2021. The recurrence of fatal accidents within a very short period was unusual. The accident type, project type, employees of the decedents, and other characteristics of these fatal accidents were identified.

As shown in Table 2, all decedents were over 40-year-old. The construction industry faces the challenge of an aging workforce. To remain competitive in the Hong Kong construction industry, older workers are becoming a critical asset. There is evidence that older construction workers have more positive attitudes than their younger counterparts [54]. However, older workers tend to have worse safety performance given that they face a higher risk of suffering severe injury or fatal accidents [55]. The present study reinforced this observation. Physical and cognitive abilities tend to deteriorate as people get older [56]. Older workers would be more vulnerable to developing physical fatigue, cognitive inability, and musculoskeletal disorders than younger workers [57–59], contributing to various health and safety-related problems. Improving the occupational safety of older construction workers demands a collaborative effort between contractors and employers to develop effective programs that accommodate older workers' age, physical, and mental health problems [55].

**Table 2.** Characteristics of the fatal accidents in the KTD project.

| | Case Number | | | | | | | |
|---|---|---|---|---|---|---|---|---|
| | **1** | **2** | **3** | **4** | **5** | **6** | **7** | **8** |
| Year | 2014 | 2020 | 2020 | 2020 | 2020 | 2021 | 2021 | 2021 |
| Month | July | June | July | July | July | April | April | April |
| Day of the week | Thursday | Saturday | Tuesday | Tuesday | Thursday | Wednesday | Sunday | Wednesday |
| Time | 11:00 | 14:00 | 9:00 | 14:30 | 12:30 | 14:00 | 9:30 | 16:00 |
| Age | 59 | 41 | 42 | 51 | 54 | 54 | 52 | 43 |
| Gender | M | M | M | M | M | M | M | M |
| Nationality | Chinese | Indonesia | Chinese | Chinese | Chinese | Chinese | Illegal worker from mainland china | Chinese |
| Length of experience in the construction industry | NA | NA | 8 years | 10 years | Over 15 years | Over 20 years | NA | Less than one year |
| Length of experience in KTD project | 32 days | 2 months | 4 months | 3 months | 1 month | 6 months | NA | 1 month |
| Type of employer | Main contractor | 2nd tier Sub-contractor | 2nd tier Sub-contractor | 2nd tier Sub-contractor | 3rd tier Sub-contractor | 2nd tier Sub-contractor | 2nd tier Sub-contractor | 3rd tier Sub-contractor |
| Type of project | Public (Cruise terminal) | Private residential | Public (Hospital) | Infrastructure | Private residential | Public residential | Private residential | Infrastructure |
| Work trade | Electrical worker | Bamboo scaffolder | Laborer | Bar bender and fixer | Piling operative | Laborer | Bamboo scaffolder | Concretor |

It was found that seven decedents had less than six months of work experience in the KTD, including the four experienced workers (also called experienced newcomers). The findings implied that inexperienced workers and experienced newcomers were vulnerable to fatal accidents. A similar observation was found in the authors' previous study [17].

Inexperienced workers were at a greater risk of construction accidents [60] because of their poorer safety perception and lower compliance with safety policies [61]. Experienced workers are more conscious of safety standards than workers with less experience [62–64]. Moreover, tacit safety knowledge gained through work experience and injury exposure could help frontline workers detect potential hazards [65]. Surprisingly, half of the decedents in the KTD had over eight years of work experience, implying that "experience does not seem to diminish accident occurrence" [66]. Wong et al. [67] found a U-shape relationship between the number of E&M accidents and length of work experience, indicating that experienced workers (over 17 years) were exposed to a higher risk of E&M accidents. Their findings implied that longer work experience would not guarantee an accident-free environment for experienced workers. Instead, some experienced workers tended to take risky behaviors [63,68] because "they are familiar with the work and know which behaviors are not dangerous" [64]. This is consistent with the concept of risk behavior compensation, which suggests that individuals would behave less cautiously in situations where they "feel" safer or more protected [69]. Given that inexperienced workers and experienced newcomers were prone to fatal accidents in the KTD, there is a pressing need for developing effective safety training programs for newcomers to increase their safety knowledge and awareness of unique features and hazardous risks associated with the new workplace.

Notably, an Indonesian and an illegal worker were involved in fatal accidents. Because of language and safety communication problems, ethnic minorities are more likely to be involved in construction accidents [70]. Ethnic minority workers are the important construction workforce in Hong Kong. More efforts should be made to enhance the ethnic minority workers' safety performance. For instance, according to Chan et al. [70,71], offering safety information and training in their native language, recruiting safety personnel from their home countries, and fostering a safety culture among them are all recommended measures. Illegal workforces are strictly prohibited in Hong Kong. The employment of an illegal worker in the KTD was considered unusual. Illegal laborers usually receive inadequate training [72] and inappropriate safety measures [73]. Even worse, they are commonly non-locals and thus may have communication barriers. All these situations pose substantial safety risks to themselves and their co-workers. The illegal worker in the present study was employed by the second-tier subcontractor engaged in a private project. Finneran and Gibb [74] pointed out that there was still a poor practice of employing illegal workforce in many smaller organizations. Government authorities should direct closer attention toward improving safety inspection on subcontractors and private projects. More surprisingly, these two fatal accidents happened during weekends. Nielsen [75] found that "black" unregistered work was usually done on weekends in the Danish construction industry. The negative impact of working on weekends on safety performance has long been recognized [76] because working on weekends usually results in fatigue in workers [77]. Working on weekends also generates unfavorable conditions, as workers may try to complete tasks as quickly as possible to take a rest [78]. A tight work schedule might also lead to working at weekends, compromising the workers' safety awareness and interfering with their ability to make safe judgments [79]. When the construction workers were under pressure due to a tight schedule, it was easier for them to select convenient unsafe behaviors that saved time or physical strength [80]. The present study showed that characteristics such as "ethnic minority/illegal workers" and "working on weekends" could create huge safety risks. Industry practitioners and government authorities should implement adequate safety training, supervision, and measures to safeguard those risky working cohorts.

It should be noted that seven decedents were engaged in subcontracting companies, and they were employed under second-tier or even third-tier contracts. In Hong Kong, around 70% of the construction work in value was subcontracted out [81]. Similarly, over 90% of construction companies are small sized in the UK, and most work under subcontract mechanisms [82]. Many studies found that the substandard practice of subcontractors poses adverse effects on the occupational health and safety of workers [83–85]

because subcontractors arguably have limited financial and resource capacities for safety improvement [86]. Apart from its inherent hurdles, the subcontractor's safety performance appeared to be more influenced by the main contractors in terms of the quality of the project scheduling and coordination and the degree of emphasis placed on safety by the main contractors [5,87]. Given the severity of fatal accidents occurring in the subcontracting companies, the main contractors should take more active roles in helping subcontractors improve safety performance by either providing safety resources or enforcing specific safety programs [88].

## 4.2. Situational Variables, Incident Sequence, and Causes of Fatal Accidents

The eight cases can be broadly divided into two accident types, fall (Cases 1, 2, 5, 7) and struck-by (Cases 3, 4, 6, 8). To be specific, the workplaces associated with the four fall accidents included stepladders (Case 1), bamboo scaffolds (Cases 2 and 7), and subsided ground (Case 5). Struck-by accidents included struck-by machine components (Case 3), reinforcement bar structure (Case 4), a pile of panels (Case 6), and concrete skip (Cases 8). The severity of fall and struck-by accidents with high lethality has long been recognized [13,19,66,89,90]. Even worse, these accidents frequently occurred in the KTD project. There is a pressing need to identify the contributory factors to avoid the reoccurrence of such unusual events. Based on Chua and Goh [33]'s MLCM, the situational variables, incident sequence, and causes of fall and struck-by accidents are illustrated in Tables 3 and 4.

### 4.2.1. Fall Accidents

Many fatal accidents occurred because of falls from scaffolding and ladders [91]. In Case 1, the accident happened when the decedent was riding on the top third rung of the stepladder. Riding on the stepladder was unsafe as it failed to maintain three points of contact with the ladder. Consequently, workers tended to lose balance on a ladder easily. Ronk et al. [92] pointed out that only 72.2% of workers would always maintain the "three points of contact" with the ladder, but 46.4% kept their hands free while climbing up and down the stepladders. In Case 1, the rungs of the stepladder were too narrow, which could not provide a firm foothold for the worker. A nylon rope was connected at the top sixth pair of opposite ladder rungs to limit the spreading angle between ladder legs. In view of this, there were no rigid ladder locks for stabilizing the ladder. Under such unsafe conditions, the decedent could not keep his balance when carrying out the conduit wiring work. As a result, he fell from the stepladder onto the floor. Although the decedent had worn a safety belt, there was no suitable anchorage point or independent lifeline to secure the safety belt. The underlying job factors of Case 1 were the failure to provide and maintain a safe system for the conduit wiring work and the adoption of improper and unsafe means of support.

**Table 3.** Situational variables, incident sequence, and causes of fall fatalities.

| Taxonomy | Sub-Taxonomy | Case 1 | Case 2 | Case 5 | Case 7 |
|---|---|---|---|---|---|
| Situational variables | Type of location | On a wooden folding stepladder | On a bamboo scaffold | On ground | On a bamboo scaffold |
| | Type of equipment/structure involved | Stepladder | Bamboo scaffold | Piling equipment | Bamboo scaffold |
| | Type of work | Electrical wiring | Erection of scaffold | Piling | Erection of scaffold |
| | Type of interacting works | NA | A metal beam was being installed vertically atop the scaffold | NA | NA |
| Type of accident | | Fall on and from a ladder Industrial and construction area | Fall on and from scaffolding Industrial and construction area | Other fall from one level to another Industrial and construction area | Fall on and from scaffolding Industrial and construction area |
| Incident sequence | Consequences | Death | Death | Death | Death |
| | Contact event | Fall from ladder | Fall from scaffold | Fall together with the piling equipment | Fall from a bamboo scaffold |
| | Breakdown event | Lost balance | Strike against the scaffold by a metal beam | The ground suddenly subsided | NA |
| Immediate causes | Substandard/unsafe conditions | Improper and unsafe means of support, i.e., wooden stepladder with no rigid ladder lock | The upper metal beam was unstably supported using undesirable welded joints | The piling equipment was not placed and operated on firm ground with sufficient load-bearing capacity | Substandard members of the bamboo scaffolds, part of standards (vertical bamboo members), and ledgers (horizontal bamboo members) of the bamboo scaffolds were missing. Moreover, there were no bracings in some inner planes of the bamboo scaffolds |
| | Substandard/unsafe behaviors | Riding on the stepladder, fail to maintain three-point contact | Working in a dangerous zone | | Not wearing any safety helmet or safety belt |

**Table 3.** *Cont.*

| Taxonomy | Sub-Taxonomy | Case 1 | Case 2 | Case 5 | Case 7 |
|---|---|---|---|---|---|
| Safety management system failures | Lack/inadequate of safety measures | Lack of safety measures for work-at-height activities, i.e., no safety net, no anchorage for the safety belt. | No precautions such as providing suitable shelters to prevent workers from being struck by any falling objects | There is no investigation or assessment to ensure the piling equipment is placed on firm ground with sufficient load-bearing capacity. | No fall protection measures and lack of independent lifeline or anchorage points for safety harness |
| | | No rigid ladder locker for stabilizing the ladder, but only a nylon rope was connected to the ladder legs | The dangerous zone was not fenced off properly, and no warning notices were displayed | | Lack of suitable working platforms with proper access and egress |
| Underlying factors | Job factors | Failure to provide and maintain a safe system for the conduit wiring work | Failure to conduct an assessment on the conditions of the welded joint of the metal beam | Inadequate supervision and inspection of the piling work | Failure to provide information, instruction, training, and supervision |
| | | | No specific risk assessment and method statement for the work and failure to provide and maintain a safe system regarding the erection of scaffold | No comprehensive site investigation before the commencement of piling work and no specific risk assessment on the piling work | No specific risk assessment and method statement for the work and failure to provide and maintain a safe system for work |
| | Organizational factors | NA | NA | NA | Employment of an illegal worker |

**Table 4.** Situational variables, incident sequence, and causes of struck-by fatalities.

| Taxonomy | Sub-Taxonomy | Case 3 | Case 4 | Case 6 | Case 8 |
|---|---|---|---|---|---|
| Situational variables | Type of location | On the loading platform of a lorry | Inside a tunnel | On the crane compartment | On ground |
| | Type of equipment/structure involved | Machine component (4 Tons) | Reinforcement bar structure | A pile of panels (600 kg) | Concrete skip |
| | Type of work | Material handling | Reinforcement Fixing | Material handling | Concreting work |
| | Type of interacting works | NA | A team of workers was working on a reinforcement bar structure | NA | NA |
| Type of accident | | Struck by thrown, projected, or falling object-Industrial and construction area | Struck by thrown, projected, or falling object-Industrial and construction area | Struck by thrown, projected, or falling object-Industrial and construction area | Struck by thrown, projected, or falling object-Industrial and construction area |
| Incident sequence | Consequences | Death | One fatality and six injuries | Death | Death |
| | Contact event | Struck by the displaced machine component | Struck by the falling reinforcement bar structure | Struck by the displaced pile of panels | Struck by the concrete skip under lifting |
| | Breakdown event | The sudden displacement of the machine component | The collapse of the reinforcement bar structure | The pile of panels under uploading displaced | The excavation suddenly tilted |
| Immediate causes | Substandard/unsafe conditions | Squatting on the loading platform of a lorry to secure a machine component that had been loaded in a slanting position on the platform | Unstable condition of the reinforcement bar structure without sufficient support | Working in the vicinity of the crane to give the operator hand signals | Poor working environment, ground condition was uneven with slopes; Poor housekeeping, impracticable to maintain unobstructed passageway |
| | Substandard/unsafe behaviors | | NA | | Working next to the lorry-mounted crane (dangerous zone) |

**Table 4.** *Cont.*

| Taxonomy | Sub-Taxonomy | Case 3 | Case 4 | Case 6 | Case 8 |
|---|---|---|---|---|---|
| Safety management system failures | Lack/inadequate of safety measures | No communication about necessary safety measures between the decedent and his co-worker for the loading operation, even it was the first time to do this work | No extra measures to stabilize the reinforcement structure; No provision of shielding, soil stabilizing treatment | NA | No segregation for separating workers from entering the working zone |
| Underlying factors | Job factors | No supervision on the loading work | An irresponsible subcontractor to execute and supervise the construction of the structure | Failure to provide instruction and supervision for the operation of the crane | Inadequate risk assessment, supervision, and monitoring of crane operation |
| | | No safe work procedures were provided as all work arrangement made at the accident scene was based on the workers' decision only | No competent safety personnel to assess the risks of the construction of the structure | Failure to provide risk assessment and method statement to the signalman | |

In Case 2, the decedent was working on the bamboo scaffolding at the façade of the building. A 9 m long, 10-ton-metal beam was installed vertically in an upright position on the topmost floor of a building under construction. The upper beam suddenly displaced and fell, striking against a bamboo scaffold being erected below the external wall of the building, resulting in the decedent's fall from 15/F to the ground with the beam together with the collapsed scaffold. Notably, this accident did not result from the decedent's unsafe acts. Rather, unsafe conditions created by others were the immediate causes. It was found that the structural fatigue of the weak welding joints of the metal beam was the cause of the displacement of the beam. This case implied that the interacting works could pose considerable risks to surrounding workers. However, under such a hazardous condition, the danger zone was not fenced off and separated properly. There were no suitable shelters provided for workers underneath the dangerous zone. The underlying job factors included two foci. First, there was a lack of comprehensive assessment of the conditions of the welded joint of the metal beam. Second, there is a lack of preventive measures to maintain a safe system regarding the erection of the scaffold. Even though the preliminary risk assessment on the erection of bamboo scaffolding had been conducted, the potential risks exposed by the interacting works nearby were not identified.

In Case 5, the decedent worked on the ground while conducting piling work. The piling equipment suddenly subsided at a 2 m depth below the ground surface, causing the decedent to fall together with the equipment. The piling equipment was not placed and operated on firm ground with sufficient load-bearing capacity, which was the immediate cause of the accident. Inadequate supervision, inspection, and risk assessment on the piling work could be the underlying causes of the accident.

In Case 7, the construction worker fell from a bamboo scaffold when erecting the scaffold at height. Some bamboo members were found warping and cracked. Moreover, part of the standards and ledgers of the bamboo scaffold were missing, and there were no bracings in some inner planes of the bamboo scaffold. These poor conditions of the bamboo scaffold implied that the worker was exposed to unsafe conditions. In accordance with the Code of Practice for Bamboo Scaffolding Safety issued by the Labour Department of Hong Kong SAR [93], bamboo members of the scaffold shall be straight and crack-free to ensure safety in the erection of scaffolds. Any defects of bamboo-like irregular knots, dry rot, gnarls, or rotten spots may affect the strength of the bamboo members. On the other hand, the decedent wore neither a safety helmet nor a safety belt, which was viewed as unsafe. However, these unsafe conditions and substandard acts were caused by the failure of the safety management system and the underlying factors. The safety measures for working at height were deemed lacking in this case. Moreover, insufficient supervision and risk assessments on working at height were not performed.

Furthermore, the decedent was found to be an illegal worker. The recruitment of illegal workers was viewed as an underlying organizational factor. Government authorities, the main contractor, and the subcontracting company shall make joint efforts to improve site inspection and supervision and prevent similar accidents in the future. In both Cases 2 and 7, the bamboo scaffold was the place of fall. The traditional construction method, which involves bamboo scaffolding, is still adopted in the Hong Kong construction industry, even though it is perceived as less safe than steel scaffolding [94]. Based on the analysis of 119 fatalities of repair, maintenance, minor alteration, and addition works in Hong Kong, Hon and Chan [19] revealed that bamboo scaffolding was the most dangerous work trade. Although the immediate causes of these two cases were different, the underlying job factors were similar. A comprehensive risk assessment on scaffolding erection was deemed lacking in both cases.

The present findings of fall accidents causation analysis were supported by Wong et al. [15] study. Wong et al. [15] used the modified HFACS framework to examine the human mistakes associated with the fall fatalities in the Hong Kong construction industry. They found that the faults in the technical environment (i.e., "the workspace or design factors that affect the actions of individuals") were the predominant preconditions of unsafe acts. Unsafe

conditions were present in all fatal falls in the present study. Wong et al. [15] indicated that inadequate supervision was one of the contributory factors, while inadequate or improper formal process (e.g., operations, procedures, and oversight) was the root organizational factor. These factors in the MLCM were classified into the underlying job factors, and they were found to be contributory factors to the four fall accidents. A lack of supervision was also the major cause of falls from roof accidents in the Singapore construction industry [95]. Unsafe conditions and acts could be influenced by inadequate supervision, whereas the lack of supervision could be attributed to improper risk assessments [6].

4.2.2. Struck-by Accidents

The decedents of all struck-by accidents (Cases 3, 4, 6, and 8) had worked in dangerous zones. In Case 3, the decedent was squatting on the loading platform of a lorry to secure a four-ton-machine component. The loaded component suddenly displaced and struck the worker, resulting in a fatal accident. It was found that the machine component had been loaded in a slanting position, which was not in a safe manner. It was surprising that there was no communication about necessary safety measures between the decedent and his co-worker for the loading operation. Effective safety communication is crucial to promote the sharing of relevant safety information, thereby improving safety knowledge and safety compliance [96]. The lack of safety communication between the decedent and his co-worker might result in the decedent ignoring the specific risks associated with the loading operation. The accident investigation also shows that no relevant information or instruction on loading and unloading of the machine component was provided to the decedent. Neither supervision on the loading work nor safe work procedures was provided with the decedent, which was considered the underlying job factors causing the fatality. Establishing a safe system of work is deemed necessary for loading work, which shall consist of a thorough risk assessment, identification of all potential hazards, definition of a details method statement, provision of adequate safety measures, and monitoring of the implementation of measures [97].

The immediate causes of Case 6 were similar to those of Case 3. The displaced objects struck the decedent working in the vicinity of the crane under uploading. It was further found that the lack of risk assessment and supervision of crane lifting operations were the underlying job factors. In Case 8, the decedent was struck against the concrete skip under lifting while a lorry-mounted crane suddenly tilted. It was observed that the ground condition was uneven with slopes, and the crane could not be stabilized. It was also found that plants and construction materials were anywhere, and such poor housekeeping made it impracticable to separate workers from entering the working zone. This case implied that the risk assessment, supervision, and monitoring of crane operations were inadequate.

In Case 4, a team of seven workers was working on a reinforcement bar structure in a tunnel under construction. The total weight of the reinforcement bar structure was approximately 22 tons. The structure then collapsed, and the workers fell together, resulting in one fatality and six injuries. It was found that there was sufficient support to stabilize the reinforcement bar structure. In addition, no tests were performed on the soil or the reinforcing rod structure before the commencement of the tunnel works. The accident investigation report further indicated that the subcontractor who executed and supervised the structure's construction was irresponsible. There was no competent safety personnel to assess the risks of the structure. These would be the underlying job factors leading to the severe accident.

Unsafe conditions were present in all struck-by accidents, indicating that the hazardous environment was a critical contributory factor to the accidents. Hinze et al. [90] found that the adverse environmental factors were presented in the 743 struck-by accidents, and the most common environmental factor was an overhead moving or falling object. Echoed to this, the present case study showed that all struck-by accidents were attributed to moving or falling objects. The proximity between objects was a major hazard factor, which was echoed in previous studies [35,90]. All the decedents worked in dangerous zones where

limited measures were taken to separate workers from danger. Congested and crowded environment was another contributor to struck-by accidents (Case 8). Because many activities on construction sites are significant sources of noise, workers might be confronted with the difficulty of recognizing hazardous surrounding environment [98]. Supervision thus becomes vital for preventing the occurrence of accidents. However, it was absent in the four struck-by accidents, which was viewed as the underlying job factor. Fass et al. [43] indicated that inadequate site and worker supervision led to struck-by accidents. Without sufficient supervision, workers might choose unsafe behaviors or make unsafe decisions (Case 3).

## 5. Recommended Preventive Measures

Accident preventative measures were gathered in the present study by a review of the pertinent accident investigation reports. A total of 22 preventive measures were divided into administrative, behavioral, and engineering controls (Table 5). Among them, four measures were strongly emphasized: "Appoint competent persons to conduct comprehensive task-specific risk assessments", "establish and implement effective safety monitoring and control systems", "formulate safe work methods and procedures", and "provide adequate safety training, information, and instructions". These measures would be helpful to correct the underlying problems.

**Table 5.** Summary of preventive measures.

| Preventive Measures | Case Number | | | | | | | | Frequency |
| --- | :-: | :-: | :-: | :-: | :-: | :-: | :-: | :-: | :-: |
| | 1 | 2 | 3 | 4 | 5 | 6 | 7 | 8 | |
| **Administrative controls** | | | | | | | | | |
| Appointing a competent person to conduct task-specific risk assessments. | | | | | | | | | 8 |
| Establishing and implementing an effective monitoring and control system. | - | | | | | | | | 8 |
| Formulating safe work methods and procedures for the work. | | | | | - | - | | | 6 |
| Appointing a professional engineer with adequate qualifications, competence, and experience to design the structure. | - | | - | | - | - | - | - | 2 |
| Ensuring that the structure is installed strictly in accordance with the specification and method statement. | - | | - | | - | - | - | - | 2 |
| Ensuring that the scaffolds are erected by trained workmen with adequate experience and suitable safety harness. | - | | - | - | - | - | | - | 2 |
| Carrying out a thorough site investigation on the ground conditions. | - | - | - | - | | - | - | - | 2 |
| Ensuring that the crane, every chain, rope, or other lifting gear has been certified in safe working order through tests and thorough examinations. | - | - | - | - | - | | - | | 2 |
| Ensuring that the crane is only operated by a person who holds a valid certificate. | - | - | - | - | - | | - | | 2 |
| Ensuring that all site personnel and workers involved are competent. | - | | - | - | - | - | - | - | 1 |
| **Engineering controls** | | | | | | | | | |
| Avoiding carrying out any work underneath the structure being erected/altered/dismantled. | - | | - | | - | - | - | - | 2 |

**Table 5.** *Cont.*

| Preventive Measures | Case Number | | | | | | | | Frequency |
|---|---|---|---|---|---|---|---|---|---|
| | 1 | 2 | 3 | 4 | 5 | 6 | 7 | 8 | |
| Ensuring piling equipment is placed and operated on firm ground with sufficient load-bearing capacity. | - | - | - | - | | - | - | | 2 |
| Ensuring that no workers/employees work or stay underneath the materials being lifted. | - | - | - | - | - | | - | | 2 |
| Where it is not reasonably practicable to fence off the lifting zones, taking effective measures, such as appointment of sufficient watch-out personnel. | - | - | - | - | - | | - | | 2 |
| Ensuring that the structure being erected is properly and securely supported. | - | | - | | - | - | - | - | 2 |
| Ensuring the vehicle platform on which the load to be handled is of sufficient length and width and the load projection should not exceed the relevant legal requirements. | - | - | | - | - | - | - | - | 1 |
| Where work underneath the structure being erected/altered/dismantled cannot be avoided, taking necessary precautions, such as providing suitable shelters to prevent workers from being struck by any falling objects. | - | | - | - | - | - | - | - | 1 |
| Provide and properly maintain suitable and adequate safe means of access to and egress from every place of work. | - | - | - | - | - | - | | - | 1 |
| Setting the mobile crane on solid ground and using suitable mat or timber blocking with area of at least three times of the outrigger's float. | - | - | - | - | - | - | - | | 1 |
| When operating close to the edge of a soil slope or an unsupported soil excavation, ensuring that a safe distance from the edge should be maintained. | - | - | - | - | - | - | - | | 1 |
| Behavioral controls | | | | | | | | | |
| Providing all workers/employees concerned with necessary safety information, instruction and training, and personal protective equipment. | | | | | | | | | 8 |
| Observing manufacturer's recommendations and instructions to ensure the stability of the crane. | - | - | - | - | - | - | - | | 1 |

One of the major underlying job factors in the present study was a lack of comprehensive risk assessment on specific tasks performed. Risk assessment on workplace sites is critical to identify hazards, notably to support decision-making in safety programs [99]. In addition to the generic assessment, task or site-specific risk assessment becomes vital for identifying additional hazards in specific situations and formulating risk controls linked with hazards [100,101]. Furthermore, the risk assessment should be conducted by personnel with sufficient knowledge and experience [33]. Traditional risk assessment is commonly conducted manually using paper works, which may result in human errors or negligence. In Hong Kong, the Digital Works Supervision System (DWSS) adoption is required in capital works contracts with pre-tender estimates exceeding $300 million HKD since 1 April 2020 [102]. Five modules are mandatory to facilitate the digital processing of the required forms and records with one central platform. They are "request for inspection/survey check form", "site diary/site record book", "site safety inspection records", "cleansing inspection checklist", and "labor return record". It is recommended that "risk assessment" be digitalized as a part of DWSS to enhance construction projects' quality and safety performance.

As an important safety measure [77,80], safety monitoring can help identify any risks arising from task execution and thus take proper and prompt corrective actions to maintain

site safety [103]. Many decedents were exposed to hazardous conditions that were not given sufficient safety monitoring and supervision. On the one hand, a more in-depth investigation of the safety climate might be helpful to address this issue, given that safety supervision is one of the critical safety climate factors [104]. On the other hand, safety supervision and monitoring have been challenged regarding their reliance on manual inspections [105]. Recently, computer vision-based recognition technologies have been developed to automatically monitor the proximal distance between humans and moving objects [98,106]. Wearable technologies have been proposed to enhance individual safety monitoring [107,108]. The adoption of these technologies may help reduce the reliance on manual inspection and enhance safety monitoring on the job sites.

In many cases, the safe work method statements (safe work procedures) were absent. For instance, in Case 1, there was a lack of a specific method statement delineating suitable safe working procedures for the conduit wiring work. In Case 4, no safe work procedures were provided for the material loading work. In Case 6, no method statement on material handling was provided to the signalman. All these underlying job factors ultimately contributed to the fatal accidents. One-third of construction accidents occur because of the absence of safe work procedures [109]. Safety work procedures that consist of specific work steps should be well developed and implemented to guide workers to easily follow proper work practices and work safely through a task from start to finish [110]. As such, safe work method statements are recommended to be incorporated into the DWSS, and thus, supervisors and workers can quickly check and follow the safe work procedures.

Adequate safety training, information, and instructions could facilitate the development of workers' safety knowledge, awareness, and attitude [53,66]. Safety training should be enhanced in the KTD to introduce specific hazards of job sites to inexperienced workers or experienced newcomers. Apart from the identified preventive measures, extra attention should be placed on the characteristics of fatal accidents, such as "aging workers", "newcomers", "ethnic minority", "illegal worker", "working on weekends", and "subcontracting companies", as discussed earlier. In general, the present case studies suggested a safety management system failure in the KTD. Overall, the main contractors and subcontractors should establish an effective safety management system consisting of risk assessment, safety inspection, safe work procedures, and effective safety training programs [111].

## 6. Conclusions and Future Studies

This study analyzed the characteristics and causes of construction accidents occurring in a large-scale development project. The KTD involves various construction works such as commercial and residential complexes, a variety of government institutions, community facilities, and associated infrastructure works. From 2020 to 2021, construction fatalities in the KTD projects occurred more frequently than usual. The fatal accidents in the KTD projects accounted for 10% and 23% of those in the construction industry in 2020 and 2021, respectively. Therefore, there is a pressing need for in-depth research that assesses the construction accidents in the KTD to enhance safety performance in the future. Qualitative approaches were utilized to examine the fatal accidents that occurred in the project from 2014 to 2021. In conclusion, "fall" and "struck by" were the most common accidents in the project. Immediate causes, failures of the safety management system, and underlying factors of these fatal accidents were analyzed based on the MLCM framework. Unsafe work conditions were the most frequent immediate causes, while and inadequate safety measures were the major of safety management system failures. The predominant underlying job factors included a lack of task-specific risk assessment, inadequate safety monitoring and supervision, a lack of safe work procedures, and inadequate safety training. Apart from this, a number of risky characteristics of fatal accidents were also identified, such as "aging workers", "newcomers", "ethnic minority", "illegal workers", "working on weekends", and "subcontracting companies". Specific efforts should be made by government authorities and practitioners to enhance the safety management of these working cohorts.

The key preventive measures are formulated accordingly based on the accident causes and risk factors identified.

- Management approach: The clients, the main contractors, and subcontractors should establish an effective safety management system, which should consist of risk assessment, safety supervision, safe work procedures, and effective safety training programs. A comprehensive risk assessment should be conducted for each specific work procedure, especially that involving working at height or nearby heavy equipment, by qualified personnel with sufficient knowledge and experience. The main contractors should take active roles in helping subcontractors improve safety performance by enhancing safety supervision and safety training. In particular, regular safety training should be offered to "aging workers", "newcomers", and "ethnic minorities". Furthermore, the project stakeholders are encouraged to share good experiences and safety management failures, cultivating a learning culture in the construction industry.
- Safety technologies: Various safety technologies could be implemented to reduce potential risks and enhance the implementation of relevant safety programs. For instance, computer vision-based recognition technologies or wearable technologies could be leveraged to monitor the proximal distance between humans and moving objects automatically. A digital documentation system could serve as a platform for managing all necessary safety-related information, thereby supporting safety management in construction projects.
- Regulatory control: Governmental safety inspection should be strictly enforced, given that "illegal workers" and "working on weekends" are two critical risk factors identified from the fatal accidents of the KTD projects.

There were several limitations of the study. First, the impact of the features of the KTD mega interfacing projects on construction safety remains unexplored. The underlying factors, such as the project nature affecting its safety performance, are unknown. Second, the root causes of the frequent recurrence of construction accidents in the KTD are not investigated. The frequent occurrence of accidents in the same project might suggest a failure of learning from the past. These study limitations might result from the use of the collected accident investigation reports. The axiom "what you can fix depends on what you can find" is typically upheld [112]. The accident causes and preventive measures were extracted from available and reliable sources. However, the aforementioned underlying issues were not discovered by the investigators. Further studies are required to carry out an in-depth survey with the industry stakeholders, including construction organizations and frontline workers, to understand why the safety performance of the KTD was poor and why relevant stakeholders failed to learn from the past.

Despite the limitations, the present study would add to the existing body of knowledge on the safety performance and accident causes of mega interfacing projects. The study also makes a methodological contribution by selecting suitable accident causation models with a detailed evaluation of the validity and reliability. To the authors' best knowledge, this is one of the first studies to adopt this methodology in construction safety research. The methods can be replicable in further similar studies. A project-level accident causation analysis would be helpful to provide important lessons to project stakeholders, which serve as a reference for engaging clients, main contractors, subcontractors, and frontline workers in pursuing excellent safety performance. The research findings also draw government authorities' attention regarding the enforcement of safety inspection and improvement of safety planning for mega interfacing projects. Learning from accidents could be implemented and promoted in the ongoing KTD projects to prevent the recurrence of similar accidents in the future.

**Author Contributions:** Conceptualization, A.P.C.C. and Y.Y.; methodology, A.P.C.C. and Y.Y.; validation, Y.Y.; formal analysis, Y.Y. and T.N.Y.C.; investigation, Y.Y. and T.N.Y.C.; resources, T.N.Y.C.; data curation, T.N.Y.C.; writing—original draft preparation, T.N.Y.C.; writing—review and editing, Y.Y. and J.M.N.; supervision, A.P.C.C. and Y.Y.; project administration, A.P.C.C. and Y.Y.; funding acquisition, A.P.C.C. All authors have read and agreed to the published version of the manuscript.

**Funding:** This project was funded by the National Natural Science Foundation of China (Project No. 71971186), from which other deliverables will be produced with different objectives/scopes but sharing common background and methodology.

**Institutional Review Board Statement:** Not applicable.

**Informed Consent Statement:** Not applicable.

**Data Availability Statement:** Restrictions apply to the availability of these data. Data was obtained from the Coroner's Court and are available from the authors with the permission of the Coroner's Court.

**Acknowledgments:** The authors are grateful to the Coroner's Court of the Judiciary of HKSAR for the death investigation reports provided. The authors also wish to acknowledge the contribution of other team members, including Carol Hon, Wen Yi, Daniel Chan, Edmond Lam, Qingwen Zhang, and Junfeng Guan.

**Conflicts of Interest:** The authors declare no conflict of interest.

## Appendix A

**Table A1.** Comparison of Causation Models Using Case 4.

| The Incident Sequence of Case 4 / Causation Model | The Structure Suddenly Collapsed, Causing the Workers to Fall Together with the Collapsing Structure | Unstable Condition of the Reinforcement Bar Structure without Sufficient Support | No Extra Measures to Stabilize the Reinforcement Structure | No Provision of Shielding, Soil Stabilizing Treatment | An Irresponsible Subcontractor to Execute and Supervise the Construction of the Structure | No Competent Safety Personnel to Assess the Risks of the Construction of the Structure |
|---|---|---|---|---|---|---|
| Modified loss causation model (MLCM) | Incident consequence | Unsafe conditions | Lack of measures | | Underlying job factors | |
| Human factor and classification system (HFACS) | NA | Hazard by others (preconditions for unsafe acts) | Planned inappropriate operations (unsafe supervision) | | Supervisory violations (unsafe supervision) | Organizational process (organizational influences) |
| Constraint-response | Incident consequence | Inappropriate site conditions | Inappropriate construction control | | Inappropriate construction control | Inappropriate construction planning |
| Mitropoulos's system model | Incident consequence | Errors in conditions | No efforts to control conditions | | Errors in management | |
| Loughborough construction accident causation model (ConCa) | NA | Local hazards (immediate factors) | | | Inadequate supervision (shaping factors) | Inadequate risk management (originating factors) |

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
