# Peer review of "Characteristics and Causes of Construction Accidents in a Large-Scale Development Project"

_sustainability, doi:10.3390/su14084449_

Round 1

Reviewer 1 Report

The research is of interest to construction safety researchers in Hong Kong.

Words in the abstract are of different font size, please check and revise.

Lines 59-68, Lines 88-99 there may be missing citations

Uncommon abbreviations like FRAM, MLCM, ConCa and AcciMap may be avoided as readers may not be able to remember all these abbreviations and need to check and read again.

Lines 134-167 may be broken to two paragraphs.

3.1, Eight fatal accidents occurred in the KTD project between 2014 and 2021 were analyzed.  Do these cases happen on the same date in the same site? Do you obtain the data from The Coroner's Court?

Eight fatal construction accidents occurred between 2014 and 2021 in the KTD. Why only those KTD were included?

Lines 302-305, There is evidence that older con- 302 struction workers have more positive attitudes than their younger counterparts [44]. 303 Despite this, the health and safety of older workers remains a considerable challenge 304 since older workers face the higher risks of suffering severe or fatal accidents [45]

Table 2, the format of the table needs to be revised according to MDPI

Lines 400-467, some headings may help to see the groupings in a clearer way.

KTD is a mega project, are there any aspects that make it has other causes of accidents that are not so common as per other previous studies?

This paper needs some more unexpected / results that highlight its originality, contribution to literature, practical and policy. These have to be highlighted in the conclusion part too.

We may need some linkage of construction safety that can be linked to sustainability.

Polish English. Cite some construction safety papers from this journal, for example:

Sustainable construction safety knowledge sharing: A partial least square structural equation modeling and a feedforward neural network approach

Reviewer 2 Report

The manuscript entitled "Characteristics and causes of construction accidents in a large-scale development project" presents an interesting study conducted on the identification and prevention of fatal accidents in construction sector. However, the paper seems to include multiple unnecessary information and the data are presented in a hard to read manner, also, the novelty of this study wasn’t presented. The paper needs minor revisions before it is processed further, some comments follow:

Abstract. Please remove the dividing terms, i.e., Methods, Results, Conclusions, from the abstract.

Introduction section

The subsection 1.1 could be reduced as multiple unnecessary information/data regarding the Kai Tak Development have been presented. Please keep only the information that are relevant for the study.

Accident causation models subsection

The presentation is long and hard to follow. A schematic representation of this subsection will significantly improve the quality of the work.

Methods Section

Please introduce a schematic representation to present the experiment roadmap. This type of data will help the readers to easily observe the highlights and particularities of the research.

Conclusion:

The conclusion section can be improved since those are far too general.

Please rewrite the conclusions in a more quantitative form.

Please improve the conclusions and present them following the main recommendations by Academia of giving the conclusions of the study by points with highlights.
